# Self-reported sick leave following a brief preventive intervention on work-related stress: a randomised controlled trial in primary health care

Anna-Maria Hultén ®,[1] Pernilla Bjerkeli,[2] Kristina Holmgren[1]

¹Department of Health and Rehabilitation, Institute of Neuroscience and Physiology, Sahlgrenska Academy, University of Gothenburg, Gothenburg, Sweden
²Department for Public Health Research, University of Skövde, Skövde, Sweden

**Correspondence to**
Anna-Maria Hultén;
anna-maria.hulten@gu.se

## ABSTRACT

**Objectives** To evaluate the effectiveness of a brief intervention about early identification of work-related stress combined with feedback at consultation with a general practitioner (GP) on the number of self-reported sick leave days.

**Design** Randomised controlled trial. Prospective analyses of self-reported sick leave data collected between November 2015 and January 2017.

**Setting** Seven primary healthcare centres in western Sweden.

**Participants** The study included 271 employed, non-sick-listed patients aged 18–64 years seeking care for mental and/or physical health complaints. Of these, 132 patients were allocated to intervention and 139 patients to control.

**Interventions** The intervention group received a brief intervention about work-related stress, including training for GPs, screening of patients' work-related stress, feedback to patients on screening results and discussion of measures at GP consultation. The control group received treatment as usual.

**Outcome measures** The number of self-reported gross sick leave days and the number of self-reported net sick leave days, thereby also considering part-time sick leave.

**Results** At 6 months' follow-up, 220/271 (81%) participants were assessed, while at 12 months' follow-up, 241/271 (89%) participants were assessed. At 6-month follow-up, 59/105 (56%) in the intervention group and 61/115 (53%) in the control group reported no sick leave. At 12-month follow-up, the corresponding numbers were 61/119 (51%) and 57/122 (47%), respectively. There were no statistically significant differences between the intervention group and the control group in the median number of self-reported gross sick leave days and the median number of self-reported net sick leave days.

**Conclusions** The brief intervention showed no effect on the numbers of self-reported sick leave days for patients seeking care at the primary healthcare centres. Other actions and new types of interventions need to be explored to address patients' perceiving of ill health due to work-related stress.

**Trial registration number** NCT02480855.

## INTRODUCTION

Work-related stress has been in focus for decades, as it is common and affects the individual and the society in multiple ways. Depression, anxiety and musculoskeletal disorders are all possible consequences of work-related stress.[1 2] Psychosocial work conditions and work-related stress also constitute risk factors for sick leave.[1] As a consequence, almost 50% of the €3 billion paid for sickness benefits in Sweden in 2018 were due to mental disorders,[3] whereof reaction to severe stress and adjustment disorders constituted half,[4] not to mention the loss of working hours and the costs for treatment and rehabilitation.

Sick leave is a common outcome measure in research. However, the relationship between spells, morbidity and health is complex since sick leave is influenced strongly by factors other than personal health.[5–7] Hence, controversy exists about how to conceptualise sick leave in research.[6] As individual, social and economic forces jointly determine absence behaviour, aspects other than work-related stress must be considered, such as attendance motivation, absence culture and sickness benefit reform.[6–8] Even so, sick leave can be

a useful measure of health status and functioning[9] and also of future sick leave and use of disability pension.[10 11] In addition, using self-reported sick leave data makes it possible to consider the first 2 weeks of absence, which are not included in the Swedish social insurance agency's register data.

### Strengths and limitations of this study

► Few previous randomised controlled trials have focused on patients' sick leave in a primary healthcare context.

► Due to the inherent complexity in clinical trials in primary healthcare practice, the statistical power of the study might have been low.

► Sick leave data are not normally distributed and non-parametric tests therefore had to be used for the analysis.

► The outcome measure (sick leave days) is complex to interpret, as it is used both as an indicator for ill health and as a tool for treatment of ill health.

Research has shown that there is a strong correlation between sick leave and work-related stress[12 13] and that early identification of persons perceiving ill health is important for preventing sick leave.[11 14] In addition, screening for interacting individual and work factors could make it possible to focus on the patient's specific problems and aid in finding suitable treatments.[15] In Sweden, primary healthcare is responsible for basic medical treatment, nursing, preventive work and rehabilitation that do not require the medical and technical resources of a hospital or other specialist skills.[16] Primary healthcare is also considered best suited for preventive work.[16] Since general practitioners (GPs) are often the first healthcare contact for persons having physical or mental health complaints and often handle cases concerning stress and work ability,[17 18] they could be a possible starting point for preventive actions concerning ill health due to work-related stress.

Commonly, GPs working at a primary healthcare centre in Sweden have access to several other healthcare professionals, such as nurses, occupational therapists, physiotherapists and social workers, sometimes organised in psychosocial teams.[19] However, the proportion of GPs is lower than for most other comparable high-income countries, as are investments in other primary care resources.[20] In addition, earlier studies have shown that GPs might not have the prerequisites needed for early identification and treatment of patients perceiving ill health due to work-related stress in order to decrease sick leave.[21–23] Therefore, a brief preventive intervention was designed using the Work Stress Questionnaire (WSQ)[24 25] as a screening tool in combination with feedback at patient–GP consultations.[26]

## METHOD

This two-armed non-blinded randomised controlled trial (RCT) was conducted at primary healthcare centres (PHCCs) located in both urban and rural areas in the region Västra Götaland in Sweden. The trial has previously been described in detail in a study protocol.[26] The primary outcome measures for the RCT, that is, the number of registered sick leave days and the number of sick leave periods during 12 months after inclusion, have previously been reported in a research article.[27] That study was based on data from a national Swedish register, whereas the present study uses self-reported data on sick leave. An important difference between the two data sources is that register data do only include information about sick leave spells that are 15 days or longer, whereas the self-reported data include all sick leave. In addition, the evaluations of secondary outcome measures

concerning healthcare treatments and prescription medication have been published in two other articles.[28 29]

## Objectives

The objective of the study was to evaluate the effectiveness of the brief intervention about early identification of work-related stress combined with feedback at GP consultation on the number of self-reported sick leave days. The overall hypothesis was that the intervention group would have fewer sick leave days during the year after the brief intervention compared with the control group. The assumptions behind this were that (1) taking part in an initial training session increased the GPs' knowledge on work-related stress, (2) filling in the WSQ raised the patients' awareness about their level of work-related stress through self-reflection, (3) receiving feedback on WSQ results increased the patients' motivation to address their work situation and (4) the combined effect of the training session, filling in the WSQ and receiving feedback constituted a basis for in-depth discussions on relevant measures at the GP–patient consultation.

The intervention concerned sick leave due to work-related stress. Hence, it was assumed that the effect of the intervention was higher for patients reporting high work-related stress or high exposure to stressors according to the WSQ. This group was therefore studied explicitly.

## Work Stress Questionnaire

The WSQ is a self-assessment questionnaire developed in a primary healthcare context[24] and specifically designed to early identify people at risk for sick leave due to work-related stress. It has a broad scope since it is not directed towards patients with a specific diagnosis. The questionnaire has a transactional perspective, as it takes the interdependence between personal and environmental work-related characteristics into account. The 21 questions included concern both psychosocial factors and the perceived stress thereof. The questions are classified into four dimensions: influence at work, indistinct organisation and conflicts, individual demands and commitment as well as work interference with leisure time.[24] In previous studies, the WSQ was found to identify work-related stress and to predict sick leave.[30 31] In addition, the test–retest reliability and face validity of the WSQ was found to be satisfying.[24 25]

## Procedure

Seven PHCCs were included in the study, of which four were public and three were privately run. Participating GPs had to be working at least 50% of the time at the PHCC. The recruitment of patients and the performance of the interventions were conducted in parallel for a period of 4–12 weeks at each PHCC from May 2015 until January 2016. Before the intervention period, the research team visited the centre to inform the staff about the study. During the intervention period, a research assistant was stationed at the PHCC to identify and recruit eligible participants, give information on the study and

administer patients' informed consent. In addition, extra personnel resources were needed to perform the training session and to administer the WSQ to the patients. Self-reported characteristics concerning sex, age, occupational class, overall health assessed with SF-36[32] and reason for consultation were collected at baseline.

## Intervention

As an initial step, the GPs randomised to intervention received a 2-hour training session including information about work-related stress, ill health and sick leave. Instructions were also given on how to use the WSQ and how to give feedback to the participants; in addition, GPs received information on healthcare professionals available for referral. Before the GP–patient consultation, each patient filled in the WSQ and questions on background characteristics. During consultation, the intervention GPs gave feedback to the patients on the WSQ results. In addition, the GP and patient conferred about and initiated preventive measures, if needed.

## Control

The GPs randomised to control were instructed to carry on as usual with their consultations and were not informed as to whether or not the patients were participating in the study. After the consultation, the control patients filled in the WSQ and gave information about background characteristics.

## Outcomes

Follow-up data on self-reported sick leave were collected at 6 and 12 months after the intervention by telephone or email. At 6 months' follow-up, the prior 3 months were reported, while at 12 months' follow-up, the prior 6 months were reported. Data for the two follow-ups were treated separately in the analysis. The self-reported sick leave data were operationalised into two outcome measures: (1) number of self-reported gross sick leave days and (2) number of self-reported net sick leave days.

In Sweden, it is possible to have part-time sick leave while working the remaining 25%, 50% or 75% of full time. In addition, the extent of the part-time sick leave can vary during a spell. For instance, it is possible to start with full time (100%) sick leave for 2 weeks and then to continue with 50% sick leave while working 50%. To be able to account for the effect of part-time sick leave in the analysis, the self-reported net days of sick leave was used as an outcome measure. Hence, working 50% part time and being on sick leave 50% for 2 days equals one net sick leave day and two gross sick leave days.

The number of gross sick leave days for each follow-up was calculated as the sum of the total number of self-reported sick leave days during the study period. The number of net sick leave days for each follow-up was calculated by multiplying the self-reported days of sick leave for each spell by the proportion of sick leave for that spell (25%, 50%, 75% or 100% of full time). The total number of net days during the study period was then summarised.

The outcome measures were based on the following request at follow-up: Define your sick leave during the latest 3 or 6 months, each period of sick leave separately (number of days with sick leave and proportion of full time with sick leave per period: 0%, 25%, 50%, 75%, 100% or varying proportion). If a participant reported varying proportions of sick leave during a spell, it was treated as 50% of full time for the entire spell.

## Target group, sample size and power

Patients eligible to participate had to be employed, non-sick-listed, 18–64 years of age and seeking care for depression, anxiety, musculoskeletal disorders, gastrointestinal, cardiovascular conditions or other potentially stress-related symptoms. Patients with 7 days' sickness absence or more during the last month were excluded as well as patients with sickness or activity benefits or ongoing pregnancy. Patients seeking care for other causes such as psychiatric conditions (eg, schizophrenia, bipolar disorder), diabetes and urinary tract infection were also excluded. The PHCCs were economically compensated for each participant recruited.

An a priori power analysis was performed for the primary outcome measure of the RCT, the number of registered sick leave days (15 days or more), with a two-sided test, a statistical significance of p value <0.05 and an 80% power. To detect at least a 15% difference between the intervention group and the control group concerning the primary outcome, during 12 months after inclusion, at least 135 participants were needed in each group.

## Randomisation and blinding

The GPs at the participating PHCCs were randomised to either the intervention group or the control group with a 1:1 allocation. Folded slips of paper with their written names were mixed in a non-transparent bowl and subsequently drawn, one at a time, to the two groups alternately by colleagues not involved in the RCT. The patients consulting the GPs were therefore automatically allocated to either group. Due to the setup of the trial, none of the parties involved were blinded after assignment to interventions. All patients received the study information provided by the research assistant, the intervention GPs received information and training before the study started and the control GPs received information about the study but no training.

## Statistical analyses

Descriptive statistics were compiled for the main baseline characteristics of the study population included in the overall sample. In addition, separate analyses were performed for the intervention group and the control group to detect any differences between the two. Pearson's $\chi^2$ test was used to test if there were any differences between the intervention group and the control group concerning these characteristics.

Outcome data were missing for some patients due to non-response at follow-up. Therefore, a comparison was

made to test whether there were differences in characteristics between patients taking part at 6 and 12 months' follow-up, respectively, and the participants at baseline. Differences in gender proportion, age and health status were tested using $\chi^2$ test. As no statistically significant differences were observed, the patients taking part at the follow-up were included in the main analysis.

Descriptive statistics were compiled for the length of the gross sick leave periods, to get an overall understanding of the distribution of sick leave. For the analysis, the variable of self-reported gross sick leave days was categorised into four levels: 0, 1–7, 8–14 and 15 days and above. These categories were based on the Swedish sickness insurance scheme[33] stating that the employer pays sick pay for up to 2 weeks, with one qualifying day. Thereafter, sickness benefits are handled by the Swedish Social Insurance Agency. From day 8 of sickness onward, a doctor's certificate is required.

For the main analysis, a comparison between the intervention and control groups was made for the gross and net numbers of sick leave days at each follow-up (6 months and 12 months, respectively). As the distribution strongly deviated from a normal distribution, medians and quartiles were used to describe the centre and the spread of the data. The Mann-Whitney U test was used to test the difference between median values of gross and net numbers of sick leave days in the control group and the intervention group.

Additional analyses were conducted on five subsamples with patients who reported high exposure to stressors. In the subgroup analysis, the Mann-Whitney U test was used to test the difference between median number of gross sick leave days in the control group and in the intervention group. The subsamples were identified based on the results from the WSQ,[24] which were defined as follows:

1. *Low influence at work* included patients' seldom or never perceiving influence at work.
2. *High stress due to indistinct organisation and conflicts* included patients perceiving their work organisation and occurring conflicts as stressful or very stressful.
3. *High stress due to individual demands and commitment* included patients perceiving their own work demands and commitment as stressful or very stressful.

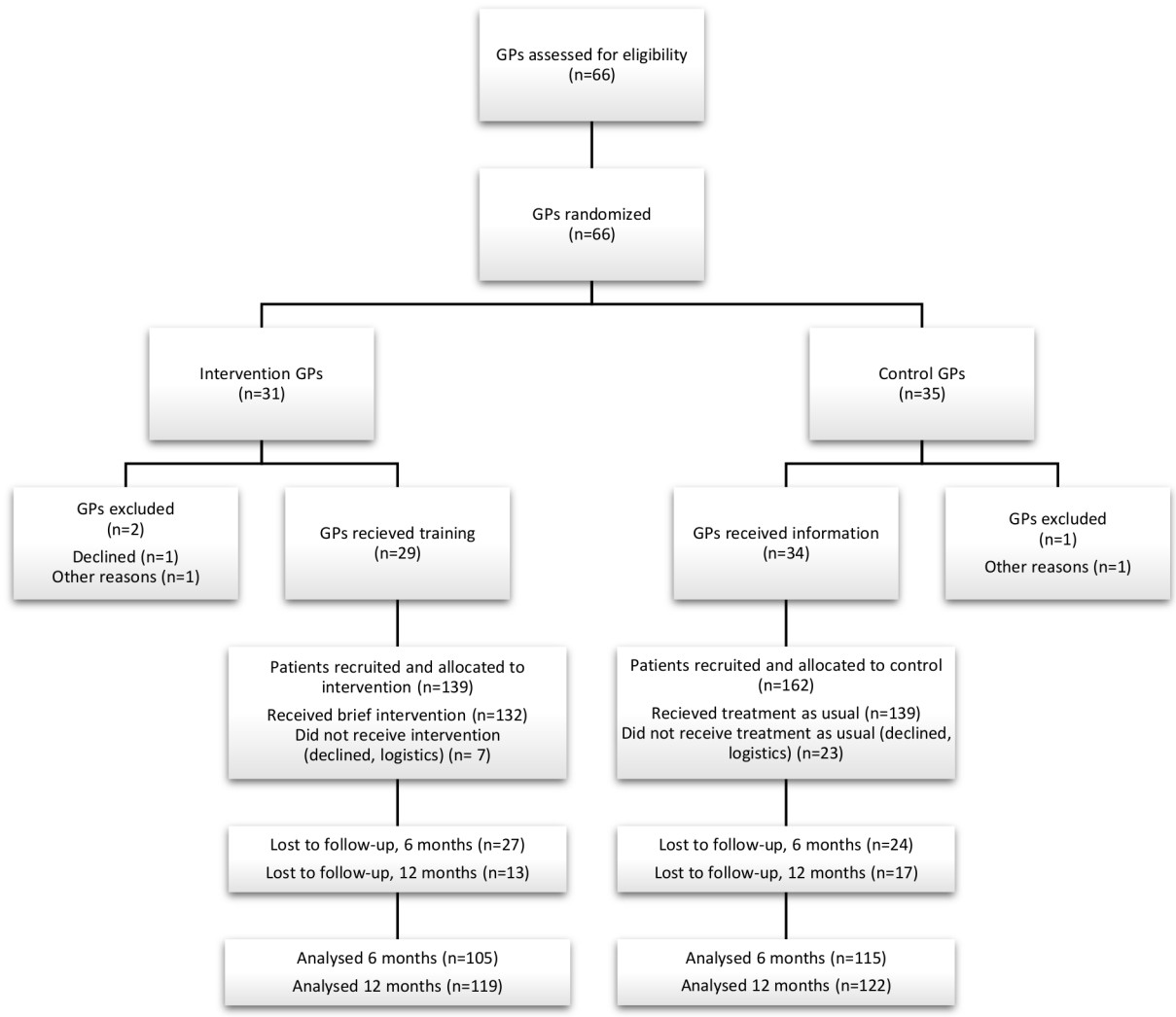

**Figure 1** Flowchart of enrolment, allocation and follow-up. GP, general practitioner

4. *High work to leisure time interference* included patients always or rather often perceiving interference between work and leisure.

5. *Effect from one subsample or more* included participants belonging to at least one of the previously described subsamples 1–4.

All answers were given on a four-point ordinal scale. A missing value in a dimension was replaced by the participant's median for that dimension, but only if there were answers to at least 50% of the questions in the dimension. The median values for each dimension were then categorised into high and low. All statistical analyses were performed in IBM SPSS Statistics 25.

## Patient and public involvement statement

There was no patient or public involvement in the planning or conduct of this trial.

## RESULTS

## Participant flow

The 66 eligible GPs at the seven PHCCs were randomised to the intervention group or the control group (figure 1). Since three GPs declined to participate or did not have patients fulfilling the criteria, there were 29 intervention GPs and 34 control GPs included. Following recruitment, 139 patients were allocated to the intervention group and 162 patients to the control group. Of these, 7 patients in the intervention group and 23 in the control group were excluded due to patients declining to participate or due to logistic reasons. Altogether, 271 patients received treatment (intervention n=132 and control n=139). Independent of group allocation, 51 of the 271 (19%) participating patients were lost to the 6-month follow-up and 30 of 271 (11%) to the 12-month follow-up. Of these, 13 (5%) did not participate in either of the follow-ups. At 6

**Table 1** Baseline characteristics of the 271 patients included in the randomised controlled trial and allocated to the intervention group or the control group

| Variable | | Total (N=271) n (%) | Intervention (N=132) n (%) | Control (N=139) n (%) | P value* |
|---|---|---|---|---|---|
| Sex | Male | 86 (32) | 44 (33) | 42 (30) | 0.582 |
| | Female | 185 (68) | 88 (67) | 97 (70) | |
| Age (years) | 18–30 | 47 (17) | 21 (16) | 26 (19) | 0.060 |
| | 31–50 | 134 (50) | 58 (44) | 76 (54) | |
| | 51–64 | 90 (33) | 53 (40) | 37 (27) | |
| Occupational class | Skilled/unskilled manual | 107 (40) | 49 (37) | 58 (42) | 0.675 |
| | Medium/low non-manual | 116 (43) | 60 (46) | 56 (40) | |
| | High-level non manual | 47 (17) | 23 (17) | 24 (17) | |
| | Missing | 1 (0) | | 1 (1) | |
| Overall health, self-rated† | Excellent/very good | 77 (28) | 34 (26) | 43 (30) | 0.526 |
| | Good | 108 (40) | 53 (40) | 55 (40) | |
| | Satisfactory/unsatisfactory | 73 (27) | 39 (30) | 34 (25) | |
| | Missing | 13 (5) | 6 (4) | 7 (5) | |
| Reason for consultation‡ | Mental or behavioural | 144 (53) | 75 (57) | 69 (50) | 0.237 |
| | Musculoskeletal | 106 (39) | 62 (47) | 44 (32) | 0.010 |
| | Gastrointestinal | 54 (20) | 26 (20) | 28 (20) | 0.927 |
| | Cardiovascular | 32 (12) | 16 (12) | 16 (13) | 0.876 |
| | Other | 56 (21) | 29 (22) | 27 (19) | 0.605 |
| WSQ results§ | Low influence at work | 108 (40) | 54 (41) | 54 (39) | 0.729 |
| | High stress organisation/conflicts | 54 (20) | 28 (21) | 26 (19) | 0.626 |
| | High stress demands/work commitment | 124 (46) | 63 (48) | 61 (44) | 0.561 |
| | High work to leisure time interference | 109 (40) | 54 (41) | 55 (40) | 0.860 |
| | Effect from one subsample or more | 188 (69) | 91 (69) | 97 (70) | 0.809 |

*Pearson's $\chi^2$ test to test differences between the intervention group and the control group.
†Short Form Health Survey, SF-36.[32]
‡More than one reason for consultation was possible.
§Work Stress Questionnaire (WSQ) results from the four dimensions dichotomised into high and low levels as well as from the summary variable including effect from at least one dimension.

months' follow-up, data from 220 patients were included in the main analysis, while at 12 months' follow-up, data from 241 patients were included. A flowchart for the enrolment, allocation and follow-ups is presented in figure 1.

## Baseline data

As shown in table 1, two-thirds of the participants (185/271) were women, 50% (134/271) were between 31 and 50 years old, and 40% (108/271) rated their health as good. The intervention group (n=132) and the control group (n=139) had similar distribution of background characteristics at baseline (n=271). However, the participants in the intervention group sought care for musculoskeletal ill health to a higher extent.

Results from the WSQ showed that 108 (40%) of the 271 participants assessed their influence at work as low, independent of group. In addition, 54 (20%) of the 271 participants reported high stress due to indistinct organisation and conflicts, while 124 (46%) reported high stress due to high individual demands and work commitment.

The fourth WSQ dimension, interference of work with leisure time, was high for 109 (40%) of the patients. Finally, 188 (69%) of the patients had stressors or stress from at least one of the four dimensions (effect from one subsample or more).

## Analysis of participants responding at follow-up

The basic characteristics of the participants in the intervention and the control groups responding at follow-up are shown in table 2. No statistically significant differences were found between baseline and responders at 6 and 12 months' follow-up concerning sex, age or self-rated health.

## Descriptive statistics of sick leave

As shown in figure 2, 59 (56%) of the 105 participants in the intervention group and 61 (53%) of the 115 participants in the control group reported no sick leave at all at the 6-month follow-up. At the 12-month follow-up, the corresponding numbers were 61 (51%) out of 119 and 57 (47%) out of 122, respectively. In addition, at 6-month

**Table 2** Characteristics of participants responding in the intervention group and the control group at 6 and 12 months' follow-up compared with baseline

| Variable, 6 months (n=220) | | Intervention | | | Control | | |
|---|---|---|---|---|---|---|---|
| | | Baseline | Follow-up* | P value† | Baseline | Follow-up* | P value† |
| Numbers | | 132 | 105 | | 139 | 115 | |
| Sex | Male | 44 | 33 | 0.756 | 42 | 35 | 0.97 |
| | Female | 88 | 72 | | 97 | 80 | |
| Age (years) | 18–30 | 21 | 17 | 0.95 | 26 | 23 | 0.908 |
| | 31–50 | 58 | 44 | | 76 | 64 | |
| | 51–64 | 53 | 44 | | 37 | 28 | |
| Overall health, self-rated‡ | Excellent/very good | 34 | 26 | 0.807 | 43 | 36 | 0.988 |
| | Good | 53 | 39 | | 55 | 45 | |
| | Satisfactory/unsatisfactory | 39 | 35 | | 34 | 27 | |
| | Missing | 6 | 5 | | 7 | 7 | |
| Variable, 12 months (n=241) | | Intervention | | | Control | | |
| | | Baseline | Follow-up§ | P value† | Baseline | Follow-up* | P value† |
| Numbers | | 132 | 119 | | 139 | 122 | |
| Sex | Male | 44 | 39 | 0.925 | 42 | 40 | 0.655 |
| | Female | 88 | 80 | | 97 | 82 | |
| Age (years) | 18–30 | 21 | 20 | 0.951 | 26 | 24 | 0.869 |
| | 31–50 | 58 | 50 | | 76 | 69 | |
| | 51–64 | 53 | 49 | | 37 | 29 | |
| Overall health, self-rated‡ | Excellent/very good | 34 | 32 | 0.968 | 43 | 38 | 0.968 |
| | Good | 53 | 46 | | 55 | 49 | |
| | Satisfactory/unsatisfactory | 39 | 35 | | 34 | 28 | |
| | Missing | 6 | 6 | | 7 | 7 | |

*6 months' follow-up.
†Testing the distribution between baseline and responders at 6 months' follow-up concerning sex, age and health with Pearson's $\chi^2$ test.
‡Short Form Health Survey, SF-36.[32]
§12 months' follow-up.

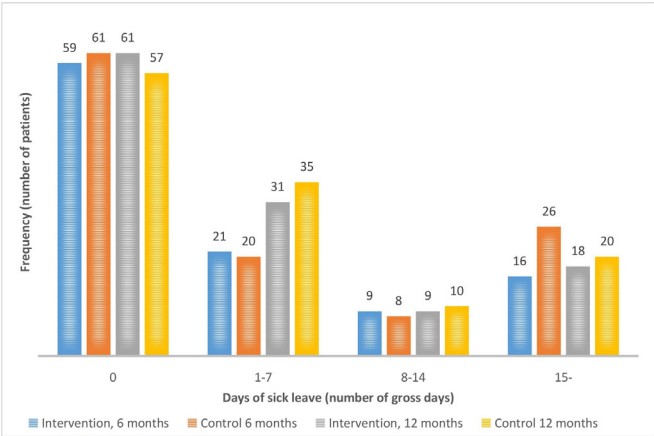

**Figure 2** Number of patients having 0, 1–7, 8–14 and ≥15 gross days of sick leave at 6 months' follow-up (n=105 in the intervention group and n=115 in the control group) and at 12 months' follow-up (n=119 in the intervention group and n=122 in the control group).

follow-up, 30 (29%) out of 105 in the intervention group and 28 (24%) out of 115 in the control group reported 1–14 days of self-reported gross sick leave (short-term sick leave). At 12-month follow-up, the corresponding numbers were 40 (34%) out of 119 in the intervention group and 45 (37%) out of 122 in the control group.

### Main analysis of sick leave

The main analysis included 220 participants at 6 months' follow-up and 241 participants at 12 months' follow-up (figure 1). As shown in table 3, the median numbers of both gross and net sick leave days at 6 months' follow-up were 0 days in the intervention group as well as in the control group. At 12 months' follow-up, the median numbers of both gross and net sick leave days were 0 days in the intervention group and 1 day in the control group. The differences were, however, not statistically significant since the p value for gross days was 0.505 and the p value for net days was 0.490.

### Sick leave in subsamples exposed to high levels of work-related stress

A comparison of the numbers of gross sick leave days for each of the five subsamples with participants who reported high levels of work related stress is shown in table 3. The differences in median number of sick leave days between the intervention group and the control group varied between 0 and 2 days in the different subsamples. In all subsamples, the median number of gross days with sick leave were equal or higher in the intervention group compared with the control group. There were, however, no statistically significant differences between the groups (p values are shown in table 3).

### DISCUSSION
### Principal findings

This study investigated differences in self-reported sick leave between patients receiving a brief intervention to prevent sick leave due to work-related stress and those

receiving treatment as usual. The results indicate that there was no significant difference in self-reported sick leave between the intervention group and the control group at 6 and 12 months' follow-up. This is in line with earlier findings from the same RCT using sick leave data from a national Swedish register including only sick leave periods 15 days and above.[27] Further, there were no significant differences in the subsamples, that is, among patients who reported high exposure to work-related stressors.

### Interpretation of findings

In this study, sick leave is used as an outcome measure, as it is considered a useful integrated measure of physical, psychological and social functioning in studies of working populations.[9] However, the relationship between ill health and sick leave is complex,[7 34] since it includes absence from work that is attributed to sickness by the employee and accepted as such by the employer[5] and other actors. To some extent, sick leave reflects employees' perceptions of their health and their behaviour in response to ill health.[9] Ill health can therefore be treated as a prerequisite of sick leave seen in relation to conditions within and outside of work.[35] Thus, previous intervention studies on sick leave have not demonstrated any effect on sick leave.[36–38] Further, short-term sick leave is considered to be more influenced by social, legal and psychological factors than health compared with long-term sick leave.[8 9] An essential component of the brief intervention was the discussion of relevant preventive measures during consultation. In general, GPs regard sickness certification as a powerful and important tool.[39] In addition, workers use sick leave as a form of self-medication and a preventive measure when perceiving strain at work.[40] Hence, the brief intervention might have contributed to GPs and patients using short-term sick leave as an early treatment and as a preventive measure to a higher extent than otherwise. Since sick leave is used both as an indicator for ill health and as a tool for treatment of ill health, an initial reduction in sick leave might not be a positive outcome of the brief intervention. This complexity might be a reason why the number of sick leave days was not lower for the intervention group than the control group.

The layout of the brief intervention is fundamental for the results retrieved. The first and perhaps foremost aspect of the intervention was to increase the GPs' knowledge and awareness about work-related stress, but the training session received might not have been exhaustive enough to raise GPs' attention to patients with work-related problems or lead them to address such a complex health issue.[41 42] Second, filling in the WSQ was expected to increase the patients' awareness about their symptoms being stress related. The use of patient-reported outcome measures has indeed been shown to improve the understanding of symptoms and facilitate communication.[43 44] However, early in the clinical reasoning process, patients could be in need of rapport building and exclusion of physical diseases and consequently resist

**Table 3** Comparison of number of sick leave days between the intervention group and the control group at 6 and 12 months' follow-up, including analysis for five subsamples

| Follow-up | Sick leave measure | Group | Number of sick leave days | | | P value‡ |
|---|---|---|---|---|---|---|
| | | | Q1* | Median | Q3† | |
| 6 months (n=220) | Gross days | Intervention | 0 | 0 | 6 | 0.449 |
| | | Control | 0 | 0 | 10 | |
| | Net days | Intervention | 0 | 0 | 5.9 | 0.398 |
| | | Control | 0 | 0 | 9 | |
| 12 months (n=241) | Gross days | Intervention | 0 | 0 | 7 | 0.505 |
| | | Control | 0 | 1 | 7.2 | |
| | Net days | Intervention | 0 | 0 | 6.2 | 0.49 |
| | | Control | 0 | 1 | 6.2 | |

| Subsamples | Sick leave measure | Group | Number of sick leave days | | | P value‡ |
|---|---|---|---|---|---|---|
| | | | Q1* | Median | Q3† | |
| Low influence | Gross days 6 months (n=89) | Intervention | 0 | 1 | 10 | 0.81 |
| | | Control | 0 | 0.5 | 27 | |
| | Gross days 12 months (n=94) | Intervention | 0 | 2 | 7 | 0.916 |
| | | Control | 0 | 2 | 6 | |
| Stress due to organisation and conflicts | Gross days 6 months (n=45) | Intervention | 0 | 0 | 7.5 | 0.931 |
| | | Control | 0 | 0 | 17.5 | |
| | Gross days 12 months (n=47) | Intervention | 0 | 2.5 | 7.7 | 0.877 |
| | | Control | 0 | 2 | 12 | |
| Stress due to commitment | Gross days 6 months (n=103) | Intervention | 0 | 1 | 14.5 | 0.793 |
| | | Control | 0 | 0 | 10.2 | |
| | Gross days 12 months (n=106) | Intervention | 0 | 2 | 8 | 0.321 |
| | | Control | 0 | 0 | 5 | |
| Work to leisure time interference | Gross days 6 months (n=89) | Intervention | 0 | 0 | 6.5 | 0.446 |
| | | Control | 0 | 0 | 30 | |
| | Gross days 12 months (n=96) | Intervention | 0 | 2 | 10 | 0.296 |
| | | Control | 0 | 0 | 5 | |
| Effect, any dimension | Gross days 6 months (n=154) | Intervention | 0 | 0 | 8 | 0.492 |
| | | Control | 0 | 0 | 19 | |
| | Gross days 12 months (n=14) | Intervention | 0 | 2 | 8.7 | 0.31 |
| | | Control | 0 | 1 | 5.7 | |

*First quartile.
†Third quartile.
‡Mann-Whitney U test.

a psychiatric explanation.[45] Third, receiving feedback on WSQ results was hypothesised to increase patients' motivation to address their work situation. However, the link between antecedents of motivation and enactment is complex. It is therefore necessary to take, for instance, past behaviour, intention, perceived behavioural control and outcome expectancy into account[46] to be able to understand this link. Thus, receiving feedback might not be sufficient to increase motivation to act. Fourth, the first three components combined in the brief intervention were assumed to constitute a basis for fruitful GP–patient discussions and initiating relevant measures. In

concordance, collaborations with patients and colleagues are seen as important elements in the referral process.[47] However, according to GPs, other aspects such as reluctance to cooperate with patients and sparse contact with colleagues could affect the referral process[47] and the measures taken. Taken together, factors related to the study setup might have diluted the effect of the intervention, so that no difference in self-reported sick leave days was detected, even for the subsamples highly exposed to stressors.

The last step of the brief intervention, that is, discussing measures, was left for the GPs to organise as they deemed

fit, rather than being specified in the study protocol. In general, GPs have a common understanding of their practice arising from the field of general practice and also from the mission of the Swedish primary healthcare system.[19] The overall way of working would therefore be similar. However, the results from a process evaluation of this RCT[48] indicate that the prerequisites for discussing measures might not have been ideal. The brief intervention was not found to assist the GPs in their work since it could alter their already well-functioning work procedure. This confirms previous findings where the use of instruments to obtain a quantitative score of depression was not perceived as useful by GPs.[49] The process evaluation also showed that the GPs could find it difficult to interpret and act on the results from the WSQ and could even question their responsibility for prevention of patients' ill health due to work-related stress, when resources were sparse. The intervention might therefore not have been efficient enough to add any effect on the days of sick leave at the follow-ups. Further, these aspects might have diminished the differences in measures taken between the intervention group and the control groups.

### Strengths and limitations

Few RCTs in primary healthcare have focused on patients' sick leave.[36–38] In some respects, this study can be considered as pragmatic since it is designed to test the impact of the brief intervention on sick leave in clinical practice. Inherent in pragmatic trials is a significant heterogeneity concerning patients, treatments and clinical settings, which leads to dilution of the effect of the intervention.[50] Consequently, pragmatic trials must be large. The initial power calculation stipulated a need for 135 individuals per group in order to detect a 15% difference between the groups. In the current study, groups with 105 and 115 participants per group at 6 months' follow-up and 119 and 122 participants per group at 12 months' follow-up were compared. The statistical power of the study is thus uncertain. It is therefore not possible to exclude the risk that there were differences between the groups that could not be detected due to lack of statistical power. However, looking more closely at the data, there are no trends that would suggest undetected differences in the main analysis. The number of days with sick leave are almost equal in the two groups regardless of outcome measure at 6 months' follow-up. At 12 months' follow-up, the median number of days is slightly higher in the control group than the intervention group, but the difference is small (0 vs 1) and not strongly reflected in the quartiles for any of the outcome variables. The subgroup analysis of individual who reported high exposure to work-related stress was performed as an attempt to focus the analysis towards a group of participants where the effect of the intervention was expected to be more pronounced, thus requiring smaller groups in order to be statistically detected. There were, however, no statically significant differences in the subgroup analysis either. It should be noted that the

non-significant differences in the subgroup analysis all point in the same direction. In all subsamples, the median number of days with sick leave is equal or higher in the intervention group than in the control group. This non-significant trend is opposite to what was detected in the main analysis where there was a slightly higher median number of days in the control group at 12 months' follow-up. The fact that none of the differences were statically significant and that the numbers point in different directions could be regarded as support for the finding that the intervention was not effective. However, the fact that all differences in the subgroup analysis pointed in the same direction could also suggest that the intervention did have an effect among those who reported high exposure to work-related stress but that the statistical power was too low to detect this difference. Another study design, including a larger group of individuals with known high exposure to stress would be needed to investigate this further.

The trial also included aspects of explanatory trials, that is, trials that aim to evaluate the efficacy of an intervention in a well-defined and controlled setting,[50] as extra personnel administered parts of the intervention. Otherwise, the study would not have been feasible. As a result, the generalisability and application in routine practice settings decreased.

The choice of outcome measures has to be taken into consideration. There are different methodological aspects and approaches to consider in using sick leave data in research.[51] Spell measures, person measures and time-based measures have to be used wisely[51] to capture any differences between the intervention group and the control group. Therefore, both the self-reported gross sick leave days and net sick leave days were used as outcome measures in this study. However, other outcome measures describing sick leave, such as number of days from intervention to sick leave and also health-related measures, might have been needed to capture an effect of the intervention.

The use of self-reported sick leave data was considered as a reasonable choice, as it made it possible to account for the first 2 weeks of sick leave. Thereby, any short periods of sick leave initiated by the workers themselves or by the GPs were included. Even so, self-reported data can be afflicted with recall bias. However, earlier studies indicate that there is good agreement between self-reported data and register information.[52 53] Even though the response rate was high, data were missing. Non-responders had to be accounted for, as this could affect the validity of trial findings.[54] Multiple imputation of missing data was not possible since the data were not normally distributed. In addition, simple imputation, such as last value carried forward, was found to be inappropriate, as it assumes a strong correlation between a prior and a later value. Since there were no statistically significant differences in characteristics between responders at baseline and at follow-up, using not imputed data for responders at 6 and 12 months' follow-up for the main analysis was considered

the best option. In addition, analysing sick leave data can be challenging, as it is not normally distributed.[50] Non-parametric tests, generally with less power, were therefore used in this study. The relatively small sample size and the statistical methods used both contributed to lowering the power. Thus, it is not possible to know whether the intervention had no effect or if it was not possible to detect an effect.

## CONCLUSIONS AND IMPLICATIONS

Based on the results from this RCT, the brief intervention showed no effect on the number of self-reported sick leave days. The study yielded information about the provision of interventions in primary healthcare. When performing RCTs in primary healthcare settings, the design is determined by what is regarded as viable. Contextual aspects such as adapted educational efforts on different levels, the patients' needs and GPs' attitudes to the intervention have to be considered thoroughly when developing and implementing interventions on preventing sick leave due to work-related stress. In addition, the results can lead to discussions about how to use sick leave as an outcome measure. Even so, there is a significant need for further research into these issues, given the individual and societal consequences of ill health due to work-related stress and the limited resources to provide treatment in a cost-effective way.

**Acknowledgements** The authors would like to acknowledge the PHCCs and especially the GPs and patients taking part in this study as well as the co-workers in the research group TIDAS for their support and feedback.

**Contributors** KH is the principal investigator and in charge of the RCT. KH was involved in designing the RCT and applying for funding. Statistical analysis was performed by A-MH in close collaboration with PB. A-MH drafted the manuscript, which was edited by KH and PB. All three authors critically reviewed and approved the final version of the manuscript.

**Funding** This study was funded by the Swedish Research Council for Health, Working Life and Welfare (2014-0936).

**Disclaimer** The Swedish Research Council for Health, Working Life and Welfare had no role in the design of the study, data collection, analysis or interpretation of data, or in writing the manuscript.

**Competing interests** None declared.

**Patient consent for publication** Not required.

**Ethics approval** The project received ethical approval, reference number 125-15, from the Regional Ethical Review Board in Gothenburg, Sweden.

**Provenance and peer review** Not commissioned; externally peer reviewed.

**Data availability statement** Data are available on reasonable request. For ethical reasons, the datasets generated and analysed during the current study are not publicly available, but they are available from the corresponding author on reasonable request.

**ORCID iD**
Anna-Maria Hultén http://orcid.org/0000-0002-4304-9459

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
