## [Reviewer comments · BMJ Open]

ARTICLE DETAILS

TITLE (PROVISIONAL)	confirm "Study protocol" if supplSelf-reported sick leave following a brief preventive intervention on work-related stress: a randomised controlled trial in primary health care
AUTHORS	Hultén, Anna-Maria; Bjerkeli, Pernilla; Holmgren, Kristina

VERSION 1 – REVIEW

REVIEWER	Stephen Stansfeld Queen Mary University of London United Kingdom I have collaborated with the research team led by Gunnel Hensing at Gothenburg University within the last 5 years.
REVIEW RETURNED	24-Jun-2020

GENERAL COMMENTS	This is an important study considering psychosocial work characteristics in the context of primary health care. There is a real need to test interventions in real world settings to reduce levels of sickness absence. Page 7/36, line 40-43: Was any assessment made of how well the GPs learnt the information on work-related stress, ill-health and sick leave? Page 7/36: Intervention: Could you give some here description of the WSQ either here or another relevant part of the methods section as this is such a crucial part of the intervention. There needs to be more than the brief description in statistical methods including some data on the reliability of the scale. Page 8/36, line 8: Do you think the control patients completing the WSQ could have influenced their behaviour in relation to sick leave? Page 8/36, paragraph 2: I don't understand the percentages in the sick leave measure. Do they refer to the proportion of time away from work during each recording period – please explain? I wonder whether it would have been advisable to separate out short and long spells of absence as there may different factors influencing short and long spells of absence. It may be the case that the intervention could influence the decision whether to take absence or not rather than the duration of absence – but maybe I have misunderstood the outcome measures? Page 14/36, Table 3: Would it have appropriate to examine sick leave days in women and men separately – or were there insufficient numbers by sex? There is an excellent discussion addressing the complexities of sickness absence research.
---

REVIEWER	Arnstein Mykletun
-----------------	-------------------

	Norwegian Institute of Public Health, Oslo & University of Tromsø, Tromsø & Haukeland University Hospital, Bergen, all Norway
REVIEW RETURNED	29-Jul-2020

GENERAL COMMENTS	I believe this study is well conducted. There is a great interest for these GP-based interventions, and in my view, too much optimism around them. This nil-finding is a valuable contribution to the literature. I think the authors should add a post-hoc power analysis indicating for example how strong the effect of the intervention would need to be to be statistically significant, for example given a two-sided test and a power of 0.80. They may want to discuss the chance of this being a type-1 error (true effect in favor of the intervention) in light of the power analysis.
---

REVIEWER	Professor Sam Leinster Norwich Medical School University of East Anglia UK
REVIEW RETURNED	30-Oct-2020

GENERAL COMMENTS	This paper is reporting one of the outcomes of a randomised trial that is registered on ClinicalTrials.gov. It is, therefore, important that the results should be published despite the finding of no effect from the intervention. There are, however, a number of areas that require clarification. 1) The criteria for selection of the participants is not stated explicitly. The reader is referred to the previously published protocol in which the inclusion and exclusion criteria are set out. However, a brief explanation of which patients were eligible for inclusion and which were excluded would remove the need to consult the protocol in order to understand the patient group involved. 2) The results include the scores on the SF36 but there is no mention of this instrument in the methodology. When did the patients complete it? 3) I do not understand how the net days of sick leave are calculated and how they differ from the gross days. What is meant by the "extent of sick leave per occasion"? 4) To what do the quartiles in Table 3 refer? 5) More emphasis should be given to the likelihood that you have a Type 2 statistical error. Your sample size calculations indicate the need for 135 participants in each group or 270 in total. Table 3 shows that the the total number of participants is 220 at 6 months and 241 at 12 months. The lack of sufficient numbers is even more marked when considering the subsamples. 5) Was any attempt made to discover the reasons for the sick leave in the two groups? It is reasonable to hypothesise that the intervention would reduce the sick leave due to mental health issues but not that due to organic illness.
---

VERSION 1 – AUTHOR RESPONSE

Reviewer 1

1. Was any assessment made of how well the GPs learnt the information on work-related stress, ill-health and sick leave?

No assessment was made of how well the information was learned. However, opinions about the training session were put forward in a focus group study concerning the intervention GPs' reasoning about using the brief intervention. This study has recently been published (Hultén, Dahlin-Ivanoff, & Holmgren, 2020). In addition, the GPs answered questions about the feasibility of the intervention and they were asked to report whether they had given feedback on the WSQ results or not during consultation (data not yet published).

2. Could you give some description of the WSQ either here or another relevant part of the methods section as this is such a crucial part of the intervention. There needs to be more than the brief description in statistical methods including some data on the reliability of the scale.

A more thorough description has been included in the Methods section (row 112-122).

3. Do you think the control patients completing the WSQ could have influenced their behaviour in relation to sick leave?

We acknowledge that we cannot entirely exclude the possibility that the behaviour of the control patients were affected by the intervention. We limited this risk by organising the data collection so that the control patients completed the WSQ after the GP consultation, whereas the intervention patients completed it before the consultation. In that way we ensured that the control patients' behaviour during the consultation was not affected by completing the questionnaire. There is, however, still a possibility that the exposure may somehow have influenced their behaviour in relation to work related stress in the time period after the consultation.

In addition, the study was organized so that the control GPs could not know whether the patient they were seeing was a control patient or a patient who was not included in the study at all. This arrangement was made to ensure that the control group received treatment as usual.

4. I don't understand the percentages in the sick leave measure. Do they refer to the proportion of time away from work during each recording period – please explain?

This is an important question and we agree that this matter needed clarification in the manuscript. It is closely linked to the design of the Swedish system for public general sickness insurance, which is arranged so that it is possible to be on part time sick leave. The percentages refer to the percentage of full time (100%) on which the participant was on sick leave during the reported sick leave spell. For example, if a participant has 50% sick leave, this means the he or she is at the same time working 50% of full time during that period. It is possible to be on sick leave for 25%, 50%, 75% or 100% of full time. Often patients who are returning to work after a long period of sick leave return gradually by first having 75% sick leave and then gradually decreasing the percentage of sick leave until they are working 100% of full time. In the calculation of "the number of net sick leave days" this has been taken into account. In the answer to the Associate Editor's comment number 1, we have included a more thorough explanation of how the number of net and gross sick leave days were operationalised (Methods section, row 154-170).

5. I wonder whether it would have been advisable to separate out short and long spells of absence as there may be different factors influencing short and long spells of absence. It may be the case that the intervention could influence the decision whether to take absence or not rather than the duration of absence – but maybe I have misunderstood the outcome measures?

This is an important question for the data analysis and the interpretation of the findings. During the analysis, different ways to classify the sick leave days were discussed. Separating short and long spells is of interest, as individual, social and economic forces jointly determine absence behaviour. Herein, regulations concerning sickness benefits is seen as an important determinant. We considered performing an analysis for those patients having one day of sick-leave or more and using 15 days of sick leave as a cut-of, since sickness benefits are handled by the Swedish Social Insurance Agency from day 15 and on. At 12 months follow-up, 58/119 participants in the intervention group and 65/122 patients in the control group had one day of sick leave or more. The power for this analysis would therefore have been lower than for the main analysis. Looking more closely at participants having 1-7 days of sick leave was also under consideration, as workers have been found to use sick leave as a form of self-medication and a preventive measure when perceiving strain at work (Danielsson, Elf, & Hensing, 2019). However, the power for this analysis would have been even lower. In addition, due to the way spells are reported, it is sometimes difficult to separate long-term and short-term sick leave spells in the data. Long-term spells may be misclassified as several consecutive short term spells which follow closely upon each other.

5. Table 3: Would it have been appropriate to examine sick leave days in women and men separately – or were there insufficient numbers by sex?

We agree that it is generally important to include stratified analysis on men and women. In the present study, we initially performed stratified analysis but we decided not to include it in the manuscript as the power in this analysis was even lower than in the main analysis (which is discussed in response to question no 1 by reviewer no 2). The results of the stratified analysis were similar to the results of the main analysis and no significant differences between intervention and control were detected.

6. There is an excellent discussion addressing the complexities of sickness absence research.

Thank you!

Reviewer 2

1. I think the authors should add a post-hoc power analysis indicating for example how strong the effect of the intervention would need to be to be statistically significant, for example given a two-sided test and a power of 0.80. They may want to discuss the chance of this being a type-1 error (true effect in favor of the intervention) in light of the power analysis.

We agree that possible lack of power is a major concern in this study. In order to emphasise this further we have included a more thorough discussion about power in the manuscript (Discussion, row 385-410). We have also included a remark about power in the conclusion of the manuscript. We have, however, not performed a post hoc power analysis. The way we interpret statistical literature, a post hoc power analysis would not necessarily add more information, as it would mainly be a backwards calculation of the p-values which could easily be misinterpreted as a true measure of the risk of committing a type 2 error (Gelman, 2019; Lenth, 2000). Instead, we have expanded our discussion of the results and of how to handle and interpret the potential lack of power in this study.

Reviewer 3

1. The criteria for selection of the participants is not stated explicitly. The reader is referred to the previously published protocol in which the inclusion and exclusion criteria are set out. However, a brief explanation of which patients were eligible for inclusion and which were excluded would remove the need to consult the protocol in order to understand the patient group involved.

Good point, a more detailed description of the criteria for inclusion and exclusion has been added (Methods section, row 173-177)

2. The results include the scores on the SF36 but there is no mention of this instrument in the methodology. When did the patients complete it?

In this study, the SF36 was only used to describe the study population at base line. Self-reported characteristics concerning variables such as overall health were collected at baseline (Methods section, row 131-133)

3. I do not understand how the net days of sick leave are calculated and how they differ from the gross days. What is meant by the "extent of sick leave per occasion"?

We agree that the description of these outcomes was not entirely clear in the previous version of the manuscript and have included an extended explanation (Methods section, row 154-170). There is also a more detailed explanation concerning the background of these outcomes in the response to the Associate Editor's comment number 1 and 2.

4. To what do the quartiles in Table 3 refer?

The quartiles refer to the number of sick leave days. Since the variables describing number of sick leave days were skewed, medians and quartiles were used to describe the centre and the spread of the data (Methods section, row 214-215). We have clarified this in Table 3.

5. More emphasis should be given to the likelihood that you have a Type 2 statistical error. Your sample size calculations indicate the need for 135 participants in each group or 270 in total. Table 3 shows that the total number of participants is 220 at 6 months and 241 at 12 months. The lack of sufficient numbers is even more marked when considering the subsamples.

We agree with the reviewer that the potential lack of power is an important concern in this study. We have added an extended discussion about this in the manuscript (Discussion, row 385-410). There is also more information concerning the power of the study in the response to reviewer 2, comment no 1.

6. Was any attempt made to discover the reasons for the sick leave in the two groups? It is reasonable to hypothesise that the intervention would reduce the sick leave due to mental health issues but not that due to organic illness.

Data about the cause for seeking care has been collected and register data on diagnosis has been retrieved. The data will be analysed and published in a future article. A decision made concerning the trial in general, was to have a wide scope and not to focus on a specific diagnosis, since ill health due to stress can take different forms. For instance, many patients seek care for physical complaints such as muscle pain.

Yours sincerely,

Anna-Maria Hultén, MSc, PhD student

References

- Bjerkeli, P. J., Skoglund, I., & Holmgren, K. (2020). Does early identification of high work related stress affect pharmacological treatment of primary care patients? - analysis of Swedish pharmacy dispensing data in a randomised control study. *BMC Fam Pract*, 21(1), 70. doi:10.1186/s12875-020-01140-x
- Danielsson, L., Elf, M., & Hensing, G. (2019). Strategies to keep working among workers with common mental disorders—a grounded theory study. *Disability and Rehabilitation*, 41(7), 786-795. doi:10.1080/09638288.2017.1408711
- Gelman, A. (2019). Don't Calculate Post-hoc Power Using Observed Estimate of Effect Size. *Annals of Surgery*, 269(1), e9-e10. doi:10.1097/SLA.0000000000002908
- Holmgren, K., Hensing, G., Bültmann, U., Hadzibajramovic, E., & Larsson, M. E. H. (2019). Does early identification of work-related stress, combined with feedback at GP-consultation, prevent sick leave in the following 12 months?: a randomized controlled trial in primary health care. *BMC Public Health*, 19(1), urn:issn:1471-2458. doi:10.1186/s12889-019-7452-3
- Hultén, A. M., Dahlin-Ivanoff, S., & Holmgren, K. (2020). Positioning work related stress - GPs' reasoning about using the WSQ combined with feedback at consultation. *BMC Family Practice*, 21(1). doi:10.1186/s12875-020-01258-y
- Lenth, R. (2000). *Two sample-size practices that I don't recommend. Comments from panel discussion*. Paper presented at the Joint Statistical Meetings, Indianapolis.
- Sandheimer, C., Hedenrud, T., Hensing, G., & Holmgren, K. (2020). Effects of a work stress intervention on healthcare use and treatment compared to treatment as usual: a randomised controlled trial in Swedish primary healthcare. *BMC Family Practice*, *In press*.

VERSION 2 – REVIEW

REVIEWER	Stephen Stansfeld Queen Mary University of London
REVIEW RETURNED	05-Jan-2021

GENERAL COMMENTS	The authors have responded well to my comments.
---

REVIEWER	Professor Sam Leinster Norwich Medical School University of East Anglia UK
REVIEW RETURNED	19-Dec-2020

GENERAL COMMENTS	Thank you for your clarifications and additions. The setting and outcomes of the study are now much easier to follow. The discussion of the reasons for the lack of effect of the intervention is convincing. The discussion of the problems with sample size is satisfactory.
--